# HPV Vaccination for Cervical Cancer Prevention in Switzerland

Emilien Jeannot [1,2,*] , Hassen Ben Abdeljelil [3] and Manuela Viviano [4]

1   Institute of Global Health, Faculty of Medicine, Geneva University, CH-1211 Geneva, Switzerland
2   Center for Excessive Gambling, Addiction Medicine (Service), Department of Psychiatry,
    Lausanne University Hospital (CHUV), Rue du Bugnon 23A, CH-1011 Lausanne, Switzerland
3   Service de Médecine Interne, Hôpital de St-Croix, Réseau Santé Balcon du Jura, Rue des Rosiers 29,
    CH-1450 Sainte Croix, Switzerland
4   Gynecology Division, Department of Obstetrics and Gynecology, Geneva University Hospital,
    CH-1211 Geneva, Switzerland
*   Correspondence: emilien.jeannot@unige.ch; Tel.: +41-22-379-464

**Definition:** Human papillomavirus (HPV) is responsible for almost all cases of cervical cancer worldwide. It is also responsible for a variety of other cancers including penile; vaginal; vulvar; anal; and oropharyngeal cancers at the base of the tongue and tonsils. There are a very large number of these HPVs, which are classified into groups from high to low risk based on their oncogenic potential. Every year in Switzerland, over 260 women develop cervical cancer, and nearly 90 of them will die from the disease. Cervical cancer affects young women and is the fourth most common cancer in women between the ages of 20 and 49 years. Among the high-risk HPV types, HPV-16 and -18 are the most common and most carcinogenic ones. Together, these two HPV types are responsible for approximately 70% of cervical cancer cases in developed countries. HPV-6 and -11 are directly responsible for 90% of genital warts. There are two effective public health interventions to prevent this cancer: screening and vaccination. The present entry provides an overview of current literature in order to present these preventative approaches and consider their use within a Swiss context. It is hoped that, going forward, this will encourage the implementation and uptake of such interventions.

**Keywords:** HPV; cervical cancer; vaccination; prevention





## 1. Contribution of This Entry

The present entry was compiled and written by the authors, at their respective institutions. The manuscript provides an overview of the current literature on HPV types, associated risks, and public health interventions in order to consider the use of prevention and intervention within the Swiss context. It is hoped that this overview will encourage further thinking from public health actors and other professionals about the implementation and use of prevention and intervention strategies.

## 2. Natural History of HPV

HPV viruses are responsible for a wide variety of skin and mucosal lesions in men and women. Similar lesions induced by viruses of the same origin are also known in the animal kingdom [1]. Cancerous lesions due to HPV viruses are found in a variety of mammals: rats, mice, rabbits, sheep, cattle, horses, deer, fallow deer, dogs, monkeys, as well as in birds and turtles.

Over the last 25 years, more than 210 genotypes of human papillomavirus have been identified. Genotypes are classified according to their tropism (skin, mucous membranes) and their oncogenic potential. Two major classes of HPV viruses have been identified [2]. Some HPVs are preferentially associated with skin lesions. HPV types 1 and 4, for example, are frequently found in warts, while HPV types 5 and 8 are linked to epidermodysplasia verruciformis [3]. This infection is relatively common, and it is estimated that it affects between 7% and 10% of the general population. This prevalence is particularly high in

school-aged children and young adults. Transmission of these viruses is facilitated by skin microtrauma that allows the virus to enter the body, as well as by contact with wet surfaces such as those found in swimming pools and gym showers [4].

HPV infects the anogenital mucosa (cervix, vulva, vagina, penis, and anus) and the oropharyngeal mucosa. Among the forty or so viruses with this tropism, some are said to be low-risk or have low oncogenic potential. This is the case with HPV-6 and -11, which are frequently found in genital warts. Others are said to be high-risk, which is the case with HPV-16 and -18 related to cervical carcinogenesis. This last group also includes HPVs known as intermediate risk: HPV 31, 33, 35, 51, etc., which are frequently found in anogenital lesions [5]. The great diversity of HPV types probably results from their evolution in different human epitheli. The mutation rate is estimated at $3.5 \times 10^6$ substitutions per site per year, which is a very high rate [6].

### 3. How Can HPV Infection Lead to Cervical Cancer?

After the onset of sexual activity, infection with one or more HPVs of different types occurs rapidly in most individuals. HPV infection is almost always transient: more than 90% of HPV infections are cleared within 1 to 2 years. In less than 10% of infected individuals, the virus persists in the mucous membranes and can lead to precancerous lesions. If left untreated, these lesions may develop into cervical, vaginal, or anal cancer. The mean time between the initial infection and the development of cervical cancer is estimated to be 20 to 30 years and at least 5 to 10 years. No specific antiviral treatment is available at this time, although research on this subject is still ongoing. Indeed, some local treatments that are less invasive than those currently proposed are being studied [7]. The most tested at present is cidofovir, a broad-spectrum antiviral that has been shown to be effective in some cases of papilloma and condyloma in clinical studies. However, it must be admitted that compared to vaccination, its efficacy appears to be low. Treatment of high-grade lesions by resection (whatever the method) allows the lesions to be removed and analysed, but these patients are known to be at high risk of local recurrence, which may require a second resection procedure, which is not without obstetrical consequences for the youngest and most common demographics of women, and is a source of concern and monitoring difficulties for all [8].

### 4. Bringing Preventive HPV Vaccines to Market

Two preventive HPV vaccines have been developed after undergoing Phase I, II, and III clinical trials. In 2006, Swissmedic authorized the marketing of Gardasil® (Merck & Co., Inc., Whitehouse Station, NJ, USA), which was marketed the following year. It is a quadrivalent vaccine against types 6, 11, 16, and 18 of the virus, the first two being responsible for most cases of condyloma acuminata.

In 2010, Cervarix® (GlaxoSmithKline Biologics, Rixensart, Belgium) was also launched. This is a bivalent vaccine against types 16 and 18 of the virus. This vaccine is rarely used and is normally no longer available in Switzerland. These vaccines are non-infectious, as they do not contain viral DNA. They are administered intramuscularly and are generally well tolerated, highly immunogenic, and with a much higher level of induced antibodies than that observed in natural infection. After four to five years, these antibody levels persist [9].

Since 2018 in Switzerland, a third nonavalent vaccine (HPV-6, -11, -16, -18, -31, -33, -45, -52, -58) is also available and is currently the most widely used given its wider vaccination coverage: Gardasil 9. It is indicated as the first choice for active immunization of individuals from the age of 11 years. In the long-term, Gardasil 9 is intended to replace Gardasil 4; however, Gardasil 4 will remain on the market as long as it is necessary to complete the vaccinations started with this vaccine. Gardasil 4, Gardasil 9 and Cervarix are not interchangeable, and the entire vaccination schedule must be completed with the same vaccine for the security and efficiency of this vaccination [10].

Since 2020, Chinese drug regulators has approved Cecolin, a generic new bivalent HPV vaccine produced by the Chinese company Innovax, but approval for marketing in the United States and European Union is unlikely [11]. However, another Indian vaccine will soon be approved for marketing by the Indian authorities and proposed for marketing outside India.

## 5. HPV Vaccination in Switzerland (Male and Female)

Since 2007, the Federal Commission for Vaccinations (FCV), initially approved vaccination for girls and women only, and then included the male sex in 2015. Vaccinated girls are gradually reaching the age of their first screening, and the challenge ahead will be to successfully reconcile these two modes of prevention, vaccination, and screening, which constitute the pillars of primary and secondary prevention against HPV [12].

Individuals aged 11-14 years are the primary target group for HPV vaccination (basic recommended vaccination is two doses at 6-month intervals). If not performed in the 11-to 14-year age group, the FOPH and the VFC recommend vaccination of females and males aged 15-19 years (catch-up vaccination, three-dose schedule at 0-, 2-, and 6-month intervals). For unvaccinated young men and women aged 20 to 26, additional vaccination can be discussed on an individual basis (three-dose regimen). The recommendation levels are based, among other things, on the disease burden and the respective usefulness of the vaccination for the different target groups [13]. Vaccination with Gardasil 9 is covered by the basic health insurance for individuals aged 11 to 26 years.

Since 2022 a new one-dose vaccination schedule has been proposed. One dose of the HPV vaccine offers protection similar to two doses for those under 21, as indicated by the World Health Organization's expert committee on immunization policy. In light of the latest data, the WHO expert committee now believes that a single dose is sufficient to protect 9- to 14-year-olds and also 15- to 20-year-olds, instead of the two previously recommended. These new recommendations should allow more girls and women to be vaccinated "while maintaining the necessary level of protection". However, this one-dose vaccination scheme has not yet been accepted in Switzerland. Vaccination is not recommended beyond the age of 26 years because the probability of having been infected with HPV once is considered very high (probability increases with age and number of sexual partners), which reduces the benefit of this vaccination for the individual. However, some studies suggest that some adults agreed after discussion with their clinician if they were not sufficiently vaccinated when they were younger (incomplete vaccination). For these adults aged 27 to 45, clinicians can discuss HPV vaccination with those who are most likely to benefit; however, if a person in this age group decides to be vaccinated, they will not be reimbursed by their health insurance [14].

Vaccines should be stored in their original packaging at a temperature between +2 °C and +8°C (ideally 5 °C) and protected from light in order to avoid damage due to freezing [15]. All vaccines are somewhat sensitive to heat or cold. Heat accelerates the decline in activity of most vaccines, reducing their shelf life. Vaccines can only be guaranteed to be effective if they are stored at the right temperature.

*Vaccination during Pregnancy*

Use of HPV vaccines is not recommended during pregnancy. As a precaution, if a person falls pregnant after having started their vaccination, the rest of the series should be postponed; however, if a dose of vaccine was given during pregnancy, no intervention is necessary. Women who are breastfeeding can receive the vaccine just as non-pregnant women [16].

## 6. Effectiveness of HPV Vaccination in Real Populations

In countries where HPV vaccination has been introduced for a long time (Australia, England) and has achieved high vaccination coverage (above 80% in the target population), new infections against which the vaccine is effective have, within 3 years, almost completely

disappeared. In women who were vaccinated before first intercourse, there was an 85-90% reduction in HPV-16/18-related high-grade dysplasia (precancerous lesions) in the cervix (CIN 2 or higher) [17,18]. Whilst studies vary in their effectiveness, all show a reduction in the prevalence of HPV covered by vaccination [18,19].

In particular, a systematic review carried out in 2022, which included 21,472 patients with cervical dysplasia clearly indicated a reduction in recurrences of CIN 1+, CIN 2+, and CIN 3 around the time of the LEEP in vaccinated patients [20].

A 2018 Cochrane review, including 26 clinical trials of a total of 73,428 participants, showed a significant reduction in the risk of precancerous lesions in vaccinated young women. This review concluded that there is strong evidence that HPV vaccines protect against precancerous cervical lesions in vaccinated adolescent girls and young people. However, protection is less effective when a portion of the population is already infected with HPV [21].

In 2018, the Australian Government stated that Australia was on track to eradicate cervical cancer and that this goal would be achieved within the next 20 years. "Australia is likely to be the first country to reach the threshold of HPV elimination," says Megan Smith, co-author of a study in the Lancet Public Health which modelled HPV eradication in Australia. No such statement or goal has been issued in Switzerland [22].

## 7. Effectiveness of HPV Vaccination in Switzerland

To date, few studies have been conducted in Switzerland to measure the effectiveness of HPV vaccination in general populations. A study conducted in the town of Geneva using self-sampling as a tool to measure HPV prevalence showed a statistically significant decrease in the prevalence of strains covered by Gardasil 4 in the study population compared to other HPV types. In this study, the authors have showed that the prevalence of 16/18 strain carriers was 7.2% in the unvaccinated subgroup of young women, whereas this prevalence was 1.1% in the vaccinated subgroup of young women ($p < 0.001$). They found the same trend in the prevalence of 6/11 infection, which was 8.3% in the unvaccinated subpopulation and only 2.1% in the vaccinated young women ($p < 0.02$). The authors did not find any difference related to age; they were statistically significant whatever the age of the participants [23].

Only one other study has been performed to estimate the effectiveness of HPV vaccination in the general population in Switzerland, which was based on patient records. Jacot-Guillarmod et al. showed that the prevalence of high-risk HPVs covered by vaccination decreased significantly (59%, $p = 0.0048$) in vaccinated participants [10]. Another example is a study that modeled the reduction in HPV prevalence in Switzerland by switching from Gardasil 4 to Gardasil 9. The results of this study showed that a gender-neutral, nonavalent vaccination program would prevent 2,979 cases of cervical cancer, 13,862 cases of CIN 3, and 15,000 cases of CIN 2, compared to a vaccination program using only Gardasil 4 in men and women over a 100-year period. These additional prevented cases of disease would correspond to a 24% decrease in the cumulative incidence of cervical cancer. This decrease would be 36% for CIN 3 and 48% for CIN 2. It would also prevent an additional 741 cervical cancer-related deaths over 100 years [24].

Finally, we should mention the CIN 3+, which aimed to examine the distribution of oncogenic HPV genotypes in women with biopsies with stage 3 cervical intraepithelial neoplasia or more severe lesions (CIN 3+) at the start of HPV vaccination programs. Up to 768 biopsies from 767 women were included in this study. Results showed that 475 (61.8%) biopsies were positive for HPV-16 and/or -18, 687 (89.5%) were positive for oncogenic genotypes of HPV-16, -18, -31, -33, -45, -52, and/or -58, and five (0.7%) were negative. There was also an extremely low vaccination coverage rate, with only 10% of women reporting having received at least one dose of the vaccine. The conclusion of this study was that in Switzerland, a potential 90% reduction in CIN 3+ lesions could be expected with the introduction of the nonavalent vaccine and a 60% reduction for the quadrivalent vaccine [25].

*Cross-Protection against Other HPV Strains after Vaccination*

To date, no study in Switzerland has demonstrated cross-protection in people vaccinated for other HPV strains not covered by the tetravalent vaccine. However, other studies on the subject have shown that such cross-protection may exist.

A study conducted in Scotland was the first to observe such cross-protection in the vaccinated population. A population-based study of women vaccinated with the bivalent vaccine at age 13 showed an 85.1% (95% confidence interval, 77.3–90.9%) reduction in HPV-31, -33, and -45. In 2018, another study, this time conducted in Holland, showed a vaccine efficacy of 61.8% (95% confidence interval, 16.7–82.5%) against strains 31, 33, and 45. The authors suggest that cross-protection has no reason to decrease over time and will persist in the vaccinated population. In Luxembourg, cross-protection against HPV 31/33/45 was also observed to have a statistically significant odds ratio (Odds Ratio = 0.41, 95% confidence interval 0.18–0.94) [26].

The most recent Dutch study on the subject showed that, for the quadrivalent vaccine, a reduction in the prevalence of strains 31 and 45 could be observed in women and men after vaccination. The most recent systematic review, which included randomized controlled trials and observational studies, concluded that there is cross-protection for strains 31 and 45 with the bivalent and quadrivalent vaccine, with at least 12 months of immunity [27]. However, current studies do not support the efficacy of this cross-protection for other HPV strains due to the frequently conflicting results found in the literature [28].

## 8. HPV Vaccine Safety and Adverse Events

Since its introduction, the Global Advisory Committee on Vaccine Safety (GACVS) and the World Health Organization (WHO), as well as state surveillance agencies monitoring the vaccine's safety and side-effects, have stated that it is safe, highly effective, and of significant public health benefit compared to the potential risks. Results of the clinical studies were significant enough for the WHO to recommend the vaccine and for most states to authorize its marketing. A second phase of surveillance of the adverse effects of the vaccine was implemented following its release.

Clinical studies evaluating the safety of theses vaccines have included more than 10,000 young women since the beginning of its development. Since 2006, 40 million doses of Gardasil 4 have been injected in the United States and more than 100 million doses in the rest of the world. International and national organizations have set up an extremely rigorous and complex system of monitoring and surveillance of the effects of theses vaccines, with collaboration between all these institutions, in order to have the most exhaustive recommendations possible on the surveillance of this vaccine [29]. In Switzerland, Swissmedic plays this role of monitoring and collecting information about adverse vaccine events.

A 2018 Cochrane review aimed to find out the risk of serious adverse events related to this vaccination. The risk of serious adverse events was comparable between the HPV vaccine and the control vaccine (a placebo or a non-HPV vaccine). The overall mortality rate is comparable (11/10,000 in the control group, 14/10,000 in the HPV vaccine group). Therefore, these HPV vaccines do not increase the risk of serious adverse events, miscarriage, or pregnancy termination. Cohort studies conducted in real-life settings to evaluate the adverse effects of HPV vaccination have not shown more adverse effects from HPV vaccination than other routine vaccinations (studies conducted in Korea and Ireland, among others). The largest study on the subject, which was performed in Australia, indicates that after 9 million doses injected nationally, syncope is the most frequently reported adverse event, with a rate of 29.6 per 100,000 doses injected. Other common adverse events observed with these vaccines are injection site reactions (approximately 60–70%) and headache (approximately 15%) [21].

*8.1. Potential Use of Self-Sampling as a Monitoring Tool for Vaccination Programs*

The use of self-sampling as an effective tool for screening for HPV and precancerous lesions is clearly presented in the scientific literature. However, its use as a tool for monitoring the effectiveness of vaccination remains to be demonstrated. The HPV impact study conducted in Geneva showed that its use was easy and well-accepted by young women. Since this research, several other studies have used self-sampling with the objective of monitoring the effectiveness of HPV vaccination in the population. Studies conducted in Canada, Italy, Australia, and Germany showed different vaccine efficacies, all while recognizing that self-sampling was a powerful tool to monitor and track HPV vaccine efficacy in the real population [17,30,31].

*8.2. How to Improve the Knowledge of Target Audiences about HPV Infection and Vaccination*

Studies conducted in Switzerland on the knowledge of health professionals regarding HPV infection and HPV vaccination showed significant gaps and lack of knowledge [32,33]. These gaps, which are found in many studies worldwide, illustrate the efforts that health professionals must make to better inform the target population on the benefits and risks of this vaccination. In recent years, there has been a significant increase in mistrust of vaccination, particularly HPV vaccination. A systematic review of 103 quantitative and qualitative studies from Europe shows that the main determinants of non-vaccination are insufficient and inadequate information about the HPV vaccination, potential side-effects of the vaccine, problems of trust in guidelines and advice given by health authorities or physicians, mistrust of a new vaccine, and perceived low vaccine efficacy. Many of these determinants could be improved by adequate communication and training of health professionals about the vaccine, including its efficacy and risks. These better-trained individuals would serve as a conduit to the target populations for conveying the right public health information [34].

The World Health Organization has made efforts in this area by providing a guide for states that includes the HPV vaccine in their vaccination programs. This good practice guidance is an aid for communicating information about the HPV vaccine. The World Health Organization guidelines clearly reiterate the importance of both epidemiological and social immunization and the crucial role of information and communication in promoting these vaccines: "Vaccination should be a social norm, for which demand and access by all members of every community is a normal and socially acceptable health behavior. The introduction of the HPV vaccine should be seen as a long-term strategy for cervical cancer prevention and communities should demand it as a social norm for their adolescent girls. This norm can be implemented through communication strategies".

The knowledge gaps we can see are often related to lack of knowledge about the virus and vaccines, which leads to reluctance or fear of vaccinating.

## 9. Prospects for HPV Vaccination with Gardasil 9 and Future Eradication

Because of its enhanced ability to protect against other HPV strains (6, 11, 16, 18, 31, 33, 45, 52, and 58), Gardasil 9 is a very powerful public health tool to drastically reduce the incidence of HPV infections. Early studies suggest that the Gardasil 9 vaccine can prevent up to 90% of cervical cancers and 96% of anal cancers worldwide. Although the distribution of oncogenic HPV prevalence varies from country to country, it can be estimated that Gardasil 9 will protect against 87.7% of cervical cancers in Asia, 91.7% in Africa, 92% in North America, 90.9% in Europe, 89.5% in Latin America and the Caribbean, and 86.5% in Australia [9].

Thirty countries have already decided to extend vaccination to young adolescent males (as young as 11 years of age), as is the case in the United States, Australia, Germany, and United Kingdom, in order to reduce the incidence of HPV-related cancers in males, which has increased since 1974. Although this vaccination has been free-of-charge in Switzerland since 1 July 2016, we have little information on vaccination coverage among young men aged 11 to 26 years, the target population for this vaccination [35].

## 10. Conclusions

Given the effectiveness of this vaccine in reducing the prevalence of oncogenic HPV strains, efforts should be made to improve knowledge among the general population. Improved information and education of the target population regarding HPV infection and the benefit of this vaccination should be reinforced in order to increase the coverage rate of this vaccine. Finally, the use of self-sampling should be part of a broader program to monitor the efficacy of vaccination, and in particular the efficacy of Gardasil 9 in Switzerland, not only among female but also among males.

**Author Contributions:** E.J., M.V. writing—original draft preparation; E.J., H.B.A., M.V. writing—review and editing. All authors have read and agreed to the published version of the manuscript.

**Funding:** This research received no external funding.

**Institutional Review Board Statement:** Not applicable.

**Informed Consent Statement:** Not applicable.

**Data Availability Statement:** Not applicable.

**Conflicts of Interest:** The authors declare no conflict of interest.

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
