# Peer review of "HPV Vaccination for Cervical Cancer Prevention in Switzerland"

_encyclopedia, doi:10.3390/encyclopedia3020036_

Round 1
Reviewer 1 Report
Well written review with up to date references. Interesting to know more about the situation in Switzerland. Minor comments only. Line 92, advice is to use same vaccine but also that it is safe to use another one. The Indian domestic vaccine has now been approved and will be used over next 3 years for girls in India. Unlike the Chinese domestic vaccine, this will be made available outside of India.
Finally there is no mention of the one dose regime which is now being adopted. I would use UK rather than Great Britain in context of HPV vaccine and screening.
Author Response
Comment 1 : Well written review with up to date references. Interesting to know more about the situation in Switzerland. Minor comments only.
Answer 1 : thank you for your support and interest in our enrty
Comment 2 : Line 92, advice is to use same vaccine but also that it is safe to use another one. The Indian domestic vaccine has now been approved and will be used over next 3 years for girls in India. Unlike the Chinese domestic vaccine, this will be made available outside of India.
Answer 2 : Thank you for these comments. We have updated our text for each of these 3 comments.
Comment 3 : Finally there is no mention of the one dose regime which is now being adopted. I would use UK rather than Great Britain in context of HPV vaccine and screening..
Answer 3 : we have added a paragraph about the new 1 dose vaccination scheme. We have also modified Great Britain by UK.
Reviewer 2 Report
This is an interesting entry generally about HPV vaccination risks and benefits, with a focus on Switzerland. I found it thorough and well written. I have no major feedback or concerns about the article. We have been looking at using HPV vaccination at the time of procedures for dysplasia; a 2021 meta-analysis by Di Donato et al in Vaccines (Basel) showed a reduction in CIN2+ for patients who had HPV vaccination around the time of the LEEP (there was only a trend towards increased clearance of the virus). It is slightly different than the rest of the discussion and the fact that is it not standard of care needs to be addressed but could be worthy or a paragraph.
Author Response
Comment 1: This is an interesting entry generally about HPV vaccination risks and benefits, with a focus on Switzerland. I found it thorough and well written. I have no major feedback or concerns about the article. We have been looking at using HPV vaccination at the time of procedures for dysplasia; a 2021 meta-analysis by Di Donato et al in Vaccines (Basel) showed a reduction in CIN2+ for patients who had HPV vaccination around the time of the LEEP (there was only a trend towards increased clearance of the virus). It is slightly different than the rest of the discussion and the fact that is it not standard of care needs to be addressed but could be worthy or a paragraph.
Answer 1 : thank you for your support and interest in our enrty. Indeed the article of Di donato in vaccine although a little away from our subject is interesting. We have added a chapter about it,
Round 2
Reviewer 1 Report
Thank you for addressing my comments
Author Response
Thank you for your comment and support
Reviewer 2 Report
I agree with the changes made.
Author Response
thank you for your comment and support